# Optimizing Detection Techniques for High-Precision Icon Recognition in Sparse Feature Spaces

## Abstract

CNNs usually work well when they can extract progressively higher-level features through the layers. In small, low-resolution images, the depth of feature extraction is limited, leading to a sparsity in the feature space. Icon detection presents a unique challenge due to the small, feature-sparse nature of the target images, which often results in limited discriminative features. To address this, we propose an icon detection model based on a Siamese network architecture. This approach draws inspiration from face recognition frameworks. The modified architectures are aimed at being well-suited for distinguishing subtle differences between icon pairs. Given the relatively sparse feature space of these icons compared to larger images, we explore several enhancements to improve performance. Key innovations include the integration of attention mechanisms to focus on informative features, multi-scale feature extraction for better detail capture and contrastive learning. These approaches lean on traditional methods of robust data augmentation to enhance performance. Additionally, we investigate dynamic margins in metric learning, adaptations for few-shot learning, and the use of Graph Neural Networks to model icon relationships. Self-supervised pretraining, adversarial training, ensemble methods, and Neural Architecture Search are employed to further refine and optimize the network. Our comprehensive evaluation demonstrates significant improvements in icon detection, highlighting the effectiveness of these advanced techniques in handling small, feature-sparse image data. This solution offers a valuable advancement in high-precision icon recognition, with potential applications in user interface design, software development, and digital asset management.

Keywords: CNN, Feature Sparse Images, Contrastive Learning, Adversarial Training, Dynamic Margins, Attention Mechanisms

## 1 Introduction

Icons are fundamental elements of modern digital assets, serving as essential markers in user interfaces, applications, and automation systems. As machines and automated processes take over routine tasks once performed by humans, the ability to accurately detect and interpret icons becomes increasingly important. This capability is crucial not only for tasks involving traditional icons but also for recognizing other small, feature-sparse images, which are common in various digital environments. While convolutional neural networks (CNNs) have achieved remarkable success in image recognition tasks, most of this progress has been focused on images rich in visual features—largely driven by datasets designed to mimic human perception. These models excel at processing large, detailed images where features such as textures, shapes, and colours are abundant. However, they are less effective when applied to smaller images, such as icons, where the visual details are limited and often sparse. This paper addresses the challenges associated with detecting and classifying small images, particularly icons. Unlike traditional models that struggle with such limited information, the proposed model is designed to generalize across different styles and contexts. This adaptability is critical in modern digital ecosystems, where personalization, brand alignment, and user-specific customization demand flexible and robust detection systems. To enhance the model's performance,

the paper explores several advanced methodologies. Self-supervised pretraining [1]is employed to enable the model to learn meaningful representations from unlabelled data, fostering improved feature extraction. The integration of attention mechanisms [2] allows the model to focus on important regions within the images, thereby enhancing its ability to differentiate similar icons effectively. Furthermore, multi-scale feature extraction [6] is utilized to ensure robustness in recognizing icons across various resolutions and sizes.

To bolster the model's resilience, adversarial training is incorporated, enabling it to withstand perturbations in the input data. The optimization process also benefits from metric learning with dynamic margins [7], refining the understanding of similarity between icons. Various loss functions—specifically cross-entropy, contrastive loss, and NT-Xent loss [1]—are compared to determine the most effective optimization strategies for this task, while adaptive margins are leveraged to fine-tune sensitivity to variations in icon features.

By developing a system optimized for smaller, feature-sparse images, this work aims to enhance the capacity for automated systems to interact with digital resources in a more intuitive and accurate way. It not only bridges the gap between current CNN-based models and small image detection but also lays the groundwork for more advanced applications in personalized automation

## 2 RELATED WORK

Pretraining techniques have been widely explored to enhance feature extraction capabilities in various image recognition tasks. One such method, SimCLR [1] (Simple Framework for Contrastive Learning of Visual Representations), has gained prominence for its self-supervised approach to pretraining. SimCLR leverages data augmentations and contrastive learning to enable models to learn powerful visual representations without the need for labeled data. Initially, SimCLR was employed in large-scale image datasets like ImageNet, focusing on rich, feature-dense images such as natural scenes and objects. However, its potential in handling small, feature-sparse images like icons has not been thoroughly explored. Our work leverages SimCLR's capacity to extract robust features from these small-scale images, using self-supervised pretraining to set up the network for downstream tasks, such as icon detection and classification.

Attention mechanisms [2] have also seen significant success in improving the performance of deep learning models on visual tasks. The Vision Transformer (ViT) [4] demonstrated the effectiveness of using self-attention to capture global dependencies in images, outperforming traditional convolutional neural networks on several image classification benchmarks. In addition to ViT, various other models have incorporated attention layers, either as an augmentation to CNNs or as part of transformer-based architectures, to enhance feature extraction from images, making attention particularly useful for images where spatial relationships are critical. In our study, we introduce attention both within and outside the Siamese network structure to assess its impact on the model's performance when dealing with small icons.

Siamese networks [3] have long been a staple in the field of metric learning, particularly for tasks like facial recognition, where the goal is to determine whether two images represent the same entity. First introduced by Bromley et al. in 1993 for signature verification tasks, the Siamese network architecture uses two identical subnetworks to compare image pairs, learning a similarity metric between them based on shared weights. This method of contrastive learning, combined with loss functions like contrastive loss, was further popularized by works in facial recognition and one-shot learning. Contrastive learning aims to minimize the distance between embeddings of similar images while maximizing the distance between dissimilar ones, making it an ideal approach for tasks involving small, feature-sparse images such as icons.

## 3 METHODOLOGY

### 3.1 SELF-SUPERVISED PRETRAINING FOR FEATURE EXTRACTION IN SMALL IMAGE DATASETS

Using SimCLR for unsupervised pretraining in your setup has several benefits, especially when dealing with small, feature-sparse images like icons. SimCLR is designed for self-supervised learning,

which means it can learn useful visual representations without requiring labelled data. The network learnt to extract features that generalize well to different images, even though no explicit labels are provided during pretraining. Once the model learned these features, it was fine-tuned on labelled data for the task of icon detection or classification. Small images like icons often contain limited visual information.

Traditional CNNs can struggle with these kinds of images because they are optimized for rich, detailed features like textures and shapes. SimCLR's approach of contrastive learning—where the network learns to differentiate between augmented views of the same image—forces the model to focus on even the subtle, distinguishing features in the image. By training the network to maximize similarity between augmented versions of the same icon and minimize similarity between different icons, SimCLR potentially enhanced feature sensitivity. Pretraining with SimCLR allows the model to learn from a larger pool of unlabelled data, which helps reduce the risk of overfitting when training the final model on a smaller labelled dataset.

## 3.2 MULTI-SCALE FEATURE EXTRACTION

The model utilizes multi-scale feature extraction, which is crucial for processing icons due to their inherent lack of distinguishable features. Icons are often minimalistic, meaning that recognizing patterns across various scales allows the network to identify the few available distinguishing characteristics, making it better suited for feature-sparse images.

Different icons can vary significantly in complexity, resolution, and style. By leveraging multi-scale extraction, the model can effectively process both simple and complex icons, leading to better generalization across different styles. Small images may have minimal distinguishing features, and these features can vary in size. For instance, one icon might exhibit very fine details, while another relies on larger, broader features.

Using multiple filter sizes (1x1, 3x3, 5x5) in parallel within the same layer enables the network to capture details at different scales. Smaller filters (1x1 or 3x3) are adept at capturing fine details like edges or textures, while larger filters (5x5 or 7x7) can identify broader patterns or shapes, which are essential for recognizing the overall structure of an icon.

## 3.3 INCORPORATING ATTENTION MECHANISMS IN FEATURE SPARSE SPACE

We incorporated an attention mechanism in the network to enhance its ability to focus on salient features within images, particularly when dealing with small, feature-sparse icons. This approach allows the model to dynamically weigh the importance of different parts of an image, improving its capacity to identify critical characteristics.

We conducted a comparison between applying attention to the embeddings generated by the Siamese network and applying attention before the generation of the final embeddings. The results empirically indicate that, in this setup, applying attention prior to embedding creation yields better accuracy.

The pre-embedding attention mechanism processes the feature maps produced by the convolutional layers, enabling the model to prioritize specific regions of the image that are more indicative of the object's identity. This strategic focus leads to embeddings that may more effectively capture the underlying structure and salient characteristics of the icons. We assume that by refining the feature extraction process before embedding generation, the network can generate richer, more informative representations, thereby enhancing overall performance in icon detection tasks

## 3.4 LOSS FUNCTION EXPLORATION

In our study, we explored different loss functions to determine the most effective approach for training the Siamese network in icon recognition tasks. The primary loss functions evaluated were Contrastive Loss, Cross-Entropy Loss, and NT-Xent Loss. Each loss function serves a different purpose and has unique characteristics that affect the model's performance

### 3.4.1 CONTRASTIVE LOSS

Contrastive Loss is particularly effective for tasks involving pairs of examples, such as in Siamese networks. This loss encourages the model to minimize the distance between embeddings of similar pairs (positive pairs) while maximizing the distance between embeddings of dissimilar pairs (negative pairs). This loss functions step the model up well when the requirement also requires information retrieval.

The contrastive loss $L$ for a pair of samples $(x_1, x_2)$ and their label $y$ is defined as:

$$L(x_1, x_2, y) = \begin{cases} \frac{1}{2}(D^2) & \text{if } y = 1 \\ \frac{1}{2}\max(0, m - D)^2 & \text{if } y = 0 \end{cases}$$

where:

- $D = \|f(x_1) - f(x_2)\|_2$ is the Euclidean distance between the feature representations $f(x_1)$ and $f(x_2)$.
- $m$ is the margin that defines how far apart the embeddings of dissimilar pairs should be.

### 3.4.2 CROSS ENTROPY LOSS

Cross-Entropy Loss is commonly used in classification tasks and measures the dissimilarity between the true distribution and the predicted distribution. While it can be applied to our task by treating the outputs as class probabilities, it does not inherently encourage distance separation in the same way that contrastive loss does. It however is more successful in cases where data is passed as pairs. And the model must discern between similar or separate classes. In case of information retrieval however, the resultant model will be used when the inference happens between pre-made pairs. This approach is practical when for one item retrieval the corresponding pool of options can be limited.

The cross-entropy loss $L$ for a single instance is defined as:

$$L(y, \hat{y}) = -\sum_{i=1}^{C} y_i \log(\hat{y}_i)$$

where:

- $C$ is the number of classes.
- $y$ is the true label (one-hot encoded vector).
- $\hat{y}$ is the predicted probability distribution (output of the model).
- $y_i$ is the true label for class $i$ (1 if the instance belongs to class $i$, 0 otherwise).
- $\hat{y}_i$ is the predicted probability for class $i$.

### 3.4.3 NT-XENT LOSS

NT-Xent (Normalized Temperature-scaled Cross-Entropy) Loss is particularly effective in contrastive learning settings. It computes the similarity between pairs of embeddings in a normalized manner, ensuring that the loss is more sensitive to the relative distances between samples. The original SimCLR paper uses this loss function to create a pretrained model. The NT-Xent loss $L$ for a pair of embeddings can be defined as:

$$L(x_i, x_j) = -\log \frac{\exp\left(\frac{\text{sim}(x_i, x_j)}{\tau}\right)}{\sum_{k=1}^{2N} \mathbb{1}_{[k \neq i]} \cdot \exp\left(\frac{\text{sim}(x_i, x_k)}{\tau}\right)}$$

where:

- $x_i$ and $x_j$ are the embeddings of the two augmented views of the same image.

- $\text{sim}(x_i, x_j)$ is the cosine similarity between the embeddings $x_i$ and $x_j$:

$$\text{sim}(x_i, x_j) = \frac{x_i \cdot x_j}{\|x_i\| \|x_j\|}$$

- $\tau$ is the temperature parameter, controlling the similarity scaling.
- The denominator sums the similarity of $x_i$ with all other embeddings, excluding itself.

### 3.4.4 LEARNABLE MARGIN

We also explored the incorporation of a learnable margin within the contrastive loss framework. The benefit of introducing a learnable margin lies in its ability to adaptively determine the optimal separation distance between positive and negative pairs during training. This flexibility allows the model to learn from the data more effectively, accommodating varying levels of similarity that may not be captured with a fixed margin. The contrastive loss $L$ with a learnable margin can be defined as:

$$L(y, d) = y \cdot \frac{1}{2} d^2 + (1 - y) \cdot \frac{1}{2} \max(0, m - d)^2$$

where:

- $y$ is a binary label indicating whether the pair is similar ($y = 1$) or dissimilar ($y = 0$).
- $d$ is the Euclidean distance between the two embeddings $x_1$ and $x_2$:

$$d = \|x_1 - x_2\|$$

- $m$ is the learnable margin parameter.

In our experiments, we found that the learnable margin-based loss function outperformed the contrastive loss in terms of accuracy metrics. This improvement highlights the importance of tailoring loss functions to the specific characteristics of the dataset and task at hand, particularly in the context of small, feature-sparse images like icons. We assume that as the model trains, the nature of the data can change, especially in small, feature-sparse images like icons. A learnable margin can help the model to respond to these changes, effectively learning which pairs are more difficult to classify and adjusting the margin accordingly. The optimal margin may vary based on the specific context of the data. For example, some icons may be more like each other than others due to their design variations. A learnable margin can incorporate this contextual information, potentially leading to improved discrimination between classes. We observed that contrastive loss requires relatively more data and long training time. Whereas cross entropy loss reached better accuracy quicker in the positive vs negative classification set up. In this paper the relative comparison is based on the model s capability to identify between positive and negative pairs, the model trains on binary cross entropy.

## 4 MODEL WORKFLOW

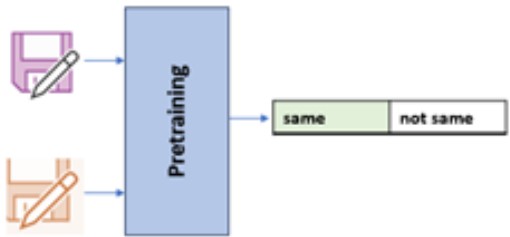

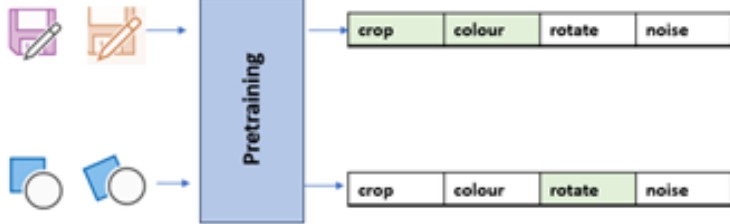

Figure 1: Pretraining: With these two exercises the network can learn to extract features from unlabelled icons. This exercise sets a strong base for the Siamese network learning

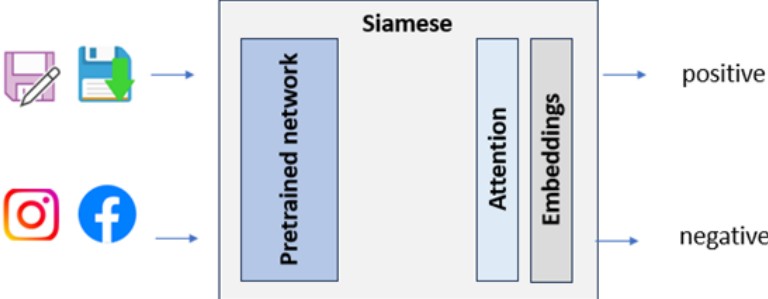

Figure 2: Model Training: The model uses the encoder from the pretrained network as the base for the Siamese network.

## 5 RESULTS

The performance of the proposed models was evaluated based on various metrics, including accuracy, recall, precision, contrastive loss, and similarity measures such as cosine similarity and Euclidean distance. The models were tested on a dataset containing icon images, where the positive pairs consisted of different representations of the same object, and the negative pairs comprised distinct objects.

### 5.1 ABLATION STUDIES

#### 5.1.1 SIAMESE NETWORK WITHOUT PRE-TRAINING

The baseline Siamese network was trained on identification of negative and positive pairs. The model demonstrated a satisfactory performance, achieving an accuracy of 81.06% on the validation dataset. However, it struggled with generalization, particularly in distinguishing between similar-looking icons.

#### 5.1.2 SIAMESE NETWORK WITH SIMCLR PRETRAINING

The incorporation of SimCLR pretraining significantly enhanced the performance of the Siamese network. The model trained with SimCLR achieved an accuracy of 88.88% on the validation dataset. The pretraining allowed the model to learn robust feature representations, which improved its ability to differentiate between similar icons. This pretraining exercise allowed the model to leverage more data which was not labelled and thus could not be incorporated in paired training.

#### 5.1.3 ATTENTION

We further evaluated the performance of the Siamese network that is already coupled with the pre-trained network when an attention was incorporated. This model achieved an accuracy of 93%, indicating that the attention mechanism positively influenced the model's focus on salient features, leading to improved recognition of icons. The exploration of attention mechanism is split in two parts viz.

a) Appling attention after the Siamese network outputs the embeddings: Applying attention after embedding allows for a reduction in the complexity of the input that the attention mechanism must process. This can lead to faster computation and less memory usage. Since the embeddings might lose some spatial context and feature details, applying attention after embedding may limit the model's ability to leverage specific spatial features in the original input data

b) Applying attention before the Siamese network output of embeddings: The model learns to focus on specific regions or aspects of the input that are most informative for the task. This can enhance feature extraction, especially for small or feature-sparse images. The model can adaptively learn which parts of the input contribute most to similarity or dissimilarity, making it potentially more robust to noise or irrelevant features in the input

### 5.2 SUMMARY OF KEY METRICS

The results indicate that incorporating SimCLR for pretraining of feature sparse small images can significantly improve the performance of Siamese networks for icon recognition tasks. It was one of the most effective enhancements we could make to boost model performance. Furthermore, the addition of attention mechanisms enhanced the models' ability to focus on crucial features. Incorporating multi scale features which is filters of different dimensions helped the model look at different levels of abstraction of the images.

Table 1: Model Performance Metrics

| Model | Dataset | Accuracy | Precision | Recall |
|---|---|---|---|---|
| No Pretraining | Training | 83.12% | 76.39% | 69.06% |
| | Validation | 81.06% | 76.81% | 77.92% |
| SimCLR Pretraining | Training | 90.04% | 93.71% | 76.26% |
| | Validation | 88.88% | 92.06% | 77.54% |
| SimCLR Pretraining + Attention outside embeddings | Training | 92.32% | 95.90% | 80.57% |
| | Validation | 91.01% | 95.83% | 79.89% |
| SimCLR Pretraining + Attention before embeddings | Training | 94.53% | 97.86% | 81.06% |
| | Validation | 93.28% | 96.44% | 79.56% |

## 6 CONCLUSION

In this study, we explored the effectiveness of various model architectures and training strategies for recognizing small, feature-sparse icons. By employing a Siamese network architecture, we demonstrated the advantages of using a dual approach that combines SimCLR for unsupervised pretraining with traditional supervised learning techniques.

Our results indicate that the SimCLR pretraining significantly enhanced feature representation, allowing the model to better differentiate between similar icons, which is critical in tasks involving minimalistic designs.

Furthermore, the incorporation of attention mechanisms resulted in notable improvements in accuracy, underscoring the importance of focusing on salient features within the images. Our experiments showed that applying attention before the embedding generation yielded better results compared to applying it afterward, suggesting that early integration of attention enhances the model's ability to capture critical information in small, sparse images.

Overall, this research highlights the potential of combining self-supervised learning methods, adaptable loss functions, and attention mechanisms to create more effective models for icon recognition. Future work may involve further optimizing these approaches and exploring additional architectures and techniques to enhance performance in diverse visual recognition tasks.

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
