# OpenReview forum: "Optimizing Detection Techniques for High-Precision Icon Recognition in Sparse Feature Spaces"
_ICLR.cc/2025/Conference — ICLR 2025 Conference Withdrawn Submission_

### Official Review · Reviewer_5f7c · 2024-10-23

**Soundness:** 2
**Presentation:** 2
**Contribution:** 1
**Rating:** 3
**Confidence:** 5

**Summary:**

In this paper, the authors proposed to explore the effectiveness of siamese networks, self-supervised pertaining, attention modules, multi-scale feature extraction, and adaptive margins in contrastive loss. The experiments have demonstrated the improvement of the proposed pertaining and attention module.

**Strengths:**

1. The task of icon recognition is pretty interesting. This type of spatial discrimination learning is worth studying.
2. This paper discusses useful technologies, like siamese network architecture, constractive pertaining, attention mechanisms, etc.
3. Experiments show the effectiveness of pertaining and attention mechanisms.

**Weaknesses:**

The completeness of the paper is poor. It lacks details on the architecture of the network and training implementation. In addition, experiments are limited. There is neither a comparison with existing methods nor an ablation comparison of designs that include multi-scale feature extraction and loss functions.

The core contribution of the paper is unclear. The authors only use the existing technologies and do not provide valuable new insights for the readers.

**Questions:**

My main concerns are listed in the weaknesses.

---

### Official Review · Reviewer_QvAU · 2024-10-28

**Soundness:** 1
**Presentation:** 1
**Contribution:** 1
**Rating:** 1
**Confidence:** 5

**Summary:**

In my opinion, this paper is not ready for ICLR.

The theoretical part of the paper lacks innovation, with simple formulas such as cross-entropy loss and cosine distance. The model structure is not innovative, the page count is less than 8 pages, there are only 8 references, and the image quality is insufficient. The experiments are not comprehensive, with only one table in the entire paper, and it only presents a very simple ablation study without comparing it to existing methods.

**Strengths:**

I found no strength

**Weaknesses:**

see summary

**Questions:**

The paper is far from ICLR paper quality.

---

### Official Review · Reviewer_MAHq · 2024-11-03

**Soundness:** 2
**Presentation:** 2
**Contribution:** 2
**Rating:** 1
**Confidence:** 4

**Summary:**

They propose an icon detection model, which is a Siamese network architecture with self-supervised pertaining, multi-scale feature extraction, and attention mechanisms. In addition, they introduce dynamic margins and various loss functions to improve performance.

**Strengths:**

1. They proposed an attention based Siamese network for icon recognition.

2. They introduced a learnable margin in the contrastive loss.

**Weaknesses:**

1. The paper is not well prepared. Some punctuation is missing, such as a period on line 161, an incorrect use of “model s” on line 252, and a missing period on line 364. Figure 1 has a resolution problem, making it blurry. In the abstract, they mention adaptations for few-shot learning and the use of GNN to model icon relationships, but I did not see either of them in the paper. There are some references in the Reference section not cited in the paper.

2. They missed a detailed introduction to the dataset. They did not try on public icon datasets, like the Icons-50 Dataset. They did not compare with other related work, like [1] and [2].

3. They lack novel contributions, all methods have been proposed by others, like SimCLR, attention-based Siamese network[3].

[1] Camilo Vargas, Qianni Zhang, and Ebroul Izquierdo. 2020. One Shot Logo Recognition Based on Siamese Neural Networks. In Proceedings of the 2020 International Conference on Multimedia Retrieval (ICMR '20). Association for Computing Machinery, New York, NY, USA, 321–325.
[2] Nakul Sharma, Abhirama Subramanyam V B Penamakuri, and Anand Mishra. 2023. Contrastive Multi-View Textual-Visual Encoding: Towards One Hundred Thousand-Scale One-Shot Logo Identification. In Proceedings of the Thirteenth Indian Conference on Computer Vision, Graphics and Image Processing (ICVGIP '22). Association for Computing Machinery, New York, NY, USA, Article 24, 1–9.
[3] Y. Liu, G. Chang, G. Fu, Y. Wei, J. Lan and J. Liu, "Self-Attention based Siamese Neural Network recognition Model," 2022 34th Chinese Control and Decision Conference (CCDC), Hefei, China, 2022, pp. 721-724

**Questions:**

1. In the dataset, you only have training and validation datasets. Where is the testing dataset?

2. I did not see the details of how you designed and implemented the attention mechanism in your model. Can you explain the details?

3. Where are the ablation study results regarding the learnable margin?

---

### Official Review · Reviewer_dv9R · 2024-11-03

**Soundness:** 1
**Presentation:** 1
**Contribution:** 1
**Rating:** 3
**Confidence:** 4

**Summary:**

This paper proposes to explore the effectiveness of multiple techniques including attention, SimCLR pretraining and learnable loss margin in the context of small and feature-sparse icon recognition. The paper finds that the following techniques are useful:
1. Using SimCLR pretraining.
2. Using attention mechanism before embedding generation.
3. Using adaptive margin loss.

**Strengths:**

1. The experimental results show an improvement compared to the baseline.
2. Icon recognition can be useful in practical settings.

**Weaknesses:**

1. Lack of novelty: This paper seems to be a combination of previously proposed method in a new dataset.
2. Inconsistency between claim and experiments: The abstract mentions that the techniques of adversarial training is included but it is not reflected in the experiments.
3. Formatting problem: Page 5 contains lots of empty lines.

**Questions:**

1. Please provide more details and experiments about the learning of margin.
2. Please provide more details about the difference between SimCLR pretraining and the Siamese network learning.
3. Please prove more details about the results of related works and dataset.

---

### Note · Authors · 2024-11-14

I have read and agree with the venue's withdrawal policy on behalf of myself and my co-authors.